# Seaweeds of the Israeli Mediterranean Sea: Nutritional and Biotechnological Potential Through Seasonal and Species Variation

**DOI:** 10.3390/md23080320

**Published:** 2025-08-04

**Authors:** Doron Yehoshua Ashkenazi, Félix L. Figueroa, Julia Vega, Shoshana Ben-Valid, Guy Paz, Eitan Salomon, Avigdor Abelson, Álvaro Israel

**Affiliations:** 1School of Zoology, Tel Aviv University, Ramat Aviv, P.O. Box 39040, Tel Aviv-Yafo 69978, Israel; shoshb@cc.huji.ac.il (S.B.-V.); avigdor@tauex.tau.ac.il (A.A.); 2Israel Oceanographic & Limnological Research, Ltd. (PBC), Tel Shikmona, P.O. Box 2336, Haifa 3102201, Israel; guy@ocean.org.il (G.P.); alvaro@ocean.org.il (Á.I.); 3Andalusian Institute for Biotechnology and Blue Development (IBYDA), Experimental Center Grice Hutchinson, Malaga University, Lomas de San Julián, 29004 Malaga, Spain; felix_lopez@uma.es (F.L.F.); juliavs90@gmail.com (J.V.); 4Israel Oceanographic & Limnological Research, National Center for Mariculture, P.O. Box 1212, Eilat 8811201, Israel; eitansol@ocean.org.il

**Keywords:** bioactive compounds, antioxidants, phenolic compounds, proteins, mycosporine-like amino acids (MAAs), seaweeds, seasonal variation, species diversity, Mediterranean sea, sustainability

## Abstract

Macroalgae (seaweeds) produce unique bioactive metabolites that have enabled their survival for millions of years, offering significant potential for human benefits. In the Israeli Mediterranean Sea, no comprehensive systematic surveys of seaweeds have been published since the 1990s, and their chemical composition remains largely unexplored. This study presents an extensive survey of intertidal seaweed communities along the shallow Israeli coastline, documenting their spatial, temporal, and biochemical diversity. Of the 320 specimens collected, 55 seaweed species were identified: 29 red (Rhodophyta), 14 brown (Phaeophyceae), and 12 green (Chlorophyta). A significant shift in species abundance was documented, with a single dominant annual bloom occurring during spring, unlike previously reported biannual blooms. Chemical analysis of the dominant species revealed significant seasonal variations in compound levels, with higher protein content in winter and increased antioxidant capacity during spring. Phenolic and natural sunscreen compounds (mycosporine-like amino acids, MAAs) showed no general seasonal trend. These findings highlight the optimal environmental conditions for seaweed growth and underscore their potential for aquaculture and biotechnology. We hypothesize that the ecologically unique conditions of the Israeli Mediterranean Sea may foster resilient seaweed species enriched with distinctive chemical properties, suitable for nutritional, health, pharmaceutical, and nutraceutical applications, particularly as climate-adaptive bioresources.

## 1. Introduction

Alongside their significant ecosystem services and nutritious ingredients, marine macroalgae (seaweeds), are among nature’s key sustainable resources, with rapidly growing human interest in their use as food, nutraceutical, cosmeceuticals, nutricosmetic, and pharmaceutical applications [1]. Seaweeds form the basis of many coastal marine ecosystems worldwide, providing shelter, nursery sites, and food for diverse marine species [2]. They are among the planet’s most productive primary producers, generating life-supporting oxygen and fixing carbon, attributes that help reduce greenhouse gases and counter global warming [3,4]. One of the most important ecosystem services of seaweeds is their unique ability to assimilate and purify dissolved nutrients and pollutants from the aquatic environment, thus helping in maintaining the ecological balance [5].

With the global population increasing, farmland decreasing, and climate change intensifying, edible seaweeds present a promising sustainable food source from the sea, potentially ensuring food security for future generations. For example, certain red seaweeds (Rhodophyta) can contain nearly 50% protein of their dry weight (DW) [6]. Generally, seaweed proteins are rich in essential amino acids, meeting FAO requirements, with species such as *Palmaria* sp., *Porphyra* sp., and *Ulva* spp. offering comparable essential amino acid levels to eggs, soybeans, and other leguminous plants [7]. Seaweeds are also a natural source of carotenoids and omega-3 fatty acids, and are the most abundant source of polysaccharides such as alginate, agar, fucoidan, ulvan, agarose, and carrageenan, used for many biomedical, cosmetic, and food applications. Some of these polysaccharides are used as dietary fibers important for a healthy digestion due to their prebiotic properties [8,9]. Furthermore, seaweeds are rich sources of vitamins A, B, C, and E, as well as essential minerals such as I, Fe, Zn, and Mg, often exhibiting concentrations 10–100 times higher than those found in land vegetables [10].

To overcome the fluctuating conditions present in the intertidal zone, seaweeds have evolved an arsenal of unique natural chemical defenses and other diverse adaptations. These adaptations include secondary metabolites that possess strong bioactive attributes such as antioxidants, phenolic compounds, and natural ultraviolet radiation (UVR) sunscreen substances such as mycosporine-like amino acids (MAAs), which we have evaluated in the current study [8,11].

Environmental oxidative stress in seaweeds can lead to the generation of reactive oxygen species (ROS), which may damage essential cellular components, including proteins, lipids, nucleic acids, and the photosynthetic apparatus. In response, seaweeds increase their antioxidant defenses, including ROS-scavenging enzymes (e.g., superoxide dismutase, catalase) and nonenzymatic secondary metabolites such as phenolic compounds, α-tocopherol (vitamin E), ascorbic acid (vitamin C), carotenoids, phycobiliproteins, and MAAs [12,13]. In the human body, ROS are associated with various clinical conditions such as aging, arthritis, strokes, atherosclerosis, diabetes, cancers, and neurodegenerative disorders. Seaweeds, with their relatively high antioxidant levels, represent a promising natural antioxidant resource, documented to mitigate oxidative damage, boost the immune system, and reduce disease risk [13].

Phenolic compounds, secondary metabolites found in terrestrial plants, algae, and many seaweed species, play vital roles in growth and survival. They are synthesized as a defense response to environmental stressors like UV radiation, metal contamination, and attacks by pathogens and insects. These compounds offer diverse beneficial biological functions, including wound healing, antioxidant, anti-inflammatory, antitumoral, antiviral, and antimicrobial properties [14].

One of the important algal photoprotective molecules are the MAAs, found largely in red seaweeds [15]. MAAs are thought to have evolved as a defense mechanism against chronic exposure to UVR in sunlight-rich, shallow-water habitats. These natural photoprotectors safeguard living cells by absorbing harmful UVR (310–362 nm range) and converting its energy into heat, thereby reducing the production of ROS [16,17]. Additionally, MAAs possess diverse bioactive properties, including potent antioxidant, anti-aging, and anti-inflammatory effects. In human skin, UVR exposure can cause photo-aging, lipid peroxidation, loss of skin resilience, wrinkles, and skin cancer. MAAs present a promising environmentally friendly natural alternative for the production of UV filters [18].

That being said, the biochemical composition of seaweeds is not stable and can vary greatly depending on environmental factors, including geographic location, seasonality, depth, irradiance, salinity, and nutrient availability, as well as on developmental stages, and biotic stressors. Currently, a significant knowledge gap remains regarding how such factors may influence the biochemical response of seaweeds, leading to the formation of specific compounds [19,20,21]. Acquiring this knowledge is especially essential in light of the rising public demands for natural, high-quality, health-promoting products that can offer an alternative to synthetic chemical materials. Currently, seaweed production is primarily focused on raw commoditized biomass, with less than 1% utilized for high-value, health-promoting products [22,23]. Moreover, out of the thousands of known seaweed species, only a relatively small number are currently being exploited. This leaves a vast underexplored biodiversity of seaweeds, offering the potential for additional utilizations [24].

The Levantine Basin, in which the Israeli Mediterranean coast is located, is environmentally unique, characterized by oligotrophic conditions and relatively high levels of temperature, salinity, and irradiance, which may encourage the development of chemically distinct seaweed species [25,26,27,28,29]. Despite the potential of seaweeds as a globally and regionally valuable natural resource, comprehensive studies examining the distribution, biodiversity, and chemodiversity of natural seaweed populations along the Israeli Mediterranean coast have remained relatively limited and infrequent, with the last systematic field survey conducted several decades ago [30,31].

In this study we had two main goals: to explore the biodiversity and seasonal variation of intertidal seaweed communities along the Israeli Mediterranean Sea coast; and to investigate the seasonal variability of the chemical constituents, including protein, polyphenol, MAA, and antioxidant levels, for the local seaweeds collected over time.

## 2. Results

### 2.1. Seaweed Survey

The distribution of red, brown, and green seaweeds across the survey collection sites and throughout the sampling period is presented in Figure 1 and Figure 2.

Red seaweeds were dominant throughout the survey for the different seasons and geographical locations (Figure 1 and Figure 2). Brown seaweeds were observed in larger numbers during winter and spring. The general number of seaweeds increased gradually with the progress of the seasons, throughout the winter period, reaching its maximal peak during spring (Figure 2). A characteristic spatial and seasonal subdivision of the intertidal euphotic zone was also observed for the different species. The most abundant species were *Jania rubens*, *Corallina elongata*, *Laurencia papillosa*, *Hypnea musciformis*, and *Gelidium pusillum* for the red seaweeds; *Padina pavonica*, *Dictyota dichotoma*, and *Sargassum vulgare* for the brown seaweeds; and all *Ulva* species (*Ulva rigida*, *Ulva compressa*, *Ulva fasciata*) and *Codium adhaerens* for the green seaweeds. The species *J. rubens*, *L. papillosa*, and *P. pavonica* were observed year-round, in all seasons (Appendix A).

A total of 55 seaweed species were identified in situ during the current survey– comprising 29 red seaweeds (Rhodophyta), 14 brown seaweeds (Heterokontophyta, class Phaeophyceae), and 12 green seaweeds (Chlorophyta), all originating primarily from the intertidal zone (Table 1).

Appendix A presents the seaweed species seasonal checklist. Appendix A presents images of all the surveyed seaweeds. As characterized for the eastern Mediterranean Levant region, the highest number of species observed during the current study belonged to the Phylum Rhodophyta, red seaweeds (53%), followed by Heterokontophyta (Phaeophyceae), brown seaweeds (25%), and Chlorophyta, green seaweeds (22%).

### 2.2. Biochemical Composition

The most dominant species from each phylum were selected for biochemical analysis, as presented in Figure 3.

#### 2.2.1. Protein Content

Except for *Hypnea musciformis* and *Codium adhaerens* which showed similar protein percentages throughout the year (Permutation Anova, *F*_3,22_ = 1.36, *p* = 0.32, One-way Anova, *F*_3,17_ = 0.9, *p* = 0.49, respectively), all other seaweeds presented significantly higher protein levels in winter compared to the other seasons (Appendix A, Table 2). Different species showed significantly different protein levels. Overall, *Dictyota dichotoma* had the highest protein levels followed by *H. musciformis* (Permutation Anova, *F*_7,178_ = 36.6, *p* < 0.0001, Table 2).

#### 2.2.2. Antioxidant Activity

For most seaweeds evaluated, a general seasonal trend was observed, showing significantly higher antioxidant capacity during spring (One-way Anova, Permutation Anova, *p* < 0.01, Appendix A, Table 2). The elevated antioxidant levels in spring were observed for all brown seaweeds and additionally for *Codium adhaerens* and *Jania rubens*. Maximal differences reached 286% for *Padina pavonica* (brown seaweeds), 170% for *C. adhaerens* (green seaweeds), and 132% for *J. rubens* (red seaweeds). *Laurencia papillosa* had the highest antioxidant levels during the summer season. Significantly lower antioxidant capacity was observed for *L. papillosa* and *U. rigida* during autumn. *Hypnea musciformis* showed similar antioxidant capacity throughout the year (Permutation Anova, *F*_3,29_ = 1.8, *p* = 0.19, Table 2). Overall, significant differences were found among the species antioxidant levels. Brown seaweeds displayed the highest antioxidant capacity, followed by the red, and green seaweed species (Permutation ANOVA, *F*_7,235_ = 109.6, *p* < 0.0001, Table 2).

#### 2.2.3. Phenolic Compounds

Phenolic compound content was evaluated for the brown and green seaweed species. Overall, no general seasonal trend was observed for the different species in their phenolic levels. *Padina pavonica* showed similar phenolic content across seasons with relatively higher content during autumn (One-way Anova, *F*_3,32_ = 2.49, *p* = 0.08, Table 2). Significant differences in the phenolic content were observed for the other seaweed species during different seasons (One-way Anova, Permutation Anova, *p* < 0.01, Appendix A, Table 2). *Sargassum vulgare* showed the highest polyphenol levels during spring (reaching a seasonal average of 37 mg PE g^−1^ DW). *Dictyota dichotoma* showed significantly higher polyphenol levels during autumn. In contrast, *Ulva rigida* had significantly low phenolic content in autumn, while *Codium adhaerens* had the lowest during winter. Generally, the brown seaweeds presented significantly higher phenolic content compared to the green seaweed species, with the highest values displayed by *S. vulgare* (Permutation Anova, *F*_4,166_ = 113.4, *p* < 0.0001, Table 2).

#### 2.2.4. Mycosporine-like Amino Acids (MAAs)

MAAs were evaluated for the red seaweeds. *Jania rubens* showed significant seasonal variation in total MAA content, reaching a peak during winter (One-way ANOVA, *F*_3,32_ = 53.8, *p* < 0.0001, Table 3), while both *Laurencia papillosa* and *Hypnea musciformis* presented similar total MAA levels across seasons (One-way ANOVA, Permutation Anova, *F*_3,32_ = 2.03, 0.11, *p* = 0.13, 0.95, respectively, Table 3). Generally, all three red seaweeds contained the MAA palythine. *J. rubens* also contained shinorine, and small amounts of asterina-330 (0.01–0.09 mg g^−1^ DW). *L. papillosa* also contained asterina-330 as a main MAA. *H. musciformis* contained in addition to palythine, large quantities of palythinol (as the main MAA), shinorine, and, during winter, also exhibited the MAA porphyra-334 (Table 3, Figure 4). A seasonal effect was also evident in the MAA composition of *J. rubens*; for example, palythine fluctuated between seasons ranging between 43 and 65%, increasing during spring. Asterina-330 altered between 7 and 20%, peaking during winter, and shinorine levels changed from 27% in spring to 47% in autumn. In contrast, *L. papillosa* and *H. musciformis* showed only small seasonal variation in their MAA composition (Table 3, Figure 4). Overall, *H. musciformis* displayed significantly higher MAA content compared to *J. rubens* and *L. papillosa* (permutation ANOVA, *F*_2,105_ = 153.3, *p* < 0.0001, Table 3).

## 3. Discussion

### 3.1. The Seaweed Survey: Ecological and Functional Insights

Overall, a total of 55 seaweed species were identified in this study out of about 300 marine macroalgae previously described for this geographical area [31]. Along the Mediterranean Sea, as one travels eastward, nutrients level decrease and seawater temperature and evaporation increase. As a result, the eastern Mediterranean Sea, particularly the Levantine Basin, represents an oligotrophic marine environment, characterized by relatively higher salinity and temperature compared to the western basin. Key nutrients, such as nitrogen and phosphorus, are generally limiting algal growth [27,29]. Seaweed biodiversity is therefore lower in the eastern Mediterranean basin than in the western basin, although it still exhibits a relatively impressive level of diversity [26,29]. Additionally, the Levantine Basin has long been exposed to increasing anthropogenic pressures, including the introduction of invasive species, pollution, and urban development [32,33].

The seaweeds inhabiting the intertidal are constantly influenced by tidal fluctuations that shape their communities. During the calm seas and low tides common in this area, seaweeds may be exposed to heavy desiccation, and high irradiance, temperature, and salinity. Conversely, in high sea conditions, turbulence and wave forces can supply the seaweeds with minerals, nutrients, and dissolved CO_2_, benefiting their growth. Biotic factors such as competition, epiphytism and herbivory, also play a fundamental part in the location and boundaries of different seaweeds [34,35]. Along the Israeli coast in particular, the dynamic environmental conditions encourage the growth of species that have developed unique physiological, biological, and chemical adaptations [36,37].

Leaf-like species such *Ulva* spp. and *Porphyra* sp. (now classified as *Neopyropia*) thrive in the upper intertidal due to their high tolerance to desiccation and their ability to maintain photosynthesis during exposure to dry conditions. *Ulva* has a worldwide distribution, is typically opportunistic with high temperature resilience, and hence can grow rapidly and spread both in summer and winter [35,38]. *Codium* spp. were also observed in the current survey as a dominant species occupying the subtidal zone, concealed beneath shaded rocky environments. *Acanthophora najadiformis* and *Hypnea musciformis* are seasonal red species and were usually observed during the colder seasons, dominating the opportunistic summer species. They were documented to be less resilient, and sensitive to high temperature, desiccation, and salinity [36]. The most dominant brown seaweed species were *Padina pavonica*, followed by *Dictyota dichotoma* and *Sargassum vulgare*. These species typically inhabit the subtidal zone. *P. pavonica* exhibited a relatively broad distribution and was observed year-round. This widespread presence is likely attributed to its ability to perform photosynthesis both when submerged and when exposed during low tides [34,36]. *Jania rubens* is a well-known prevalent red opportunistic species, and is probably the most common red seaweed in the eastern Mediterranean, occupying the intertidal and subtidal zones. Acting also as an epiphyte upon other seaweeds, it was observed year-round in the current survey and is usually closely accompanied by another dominant species - *Corallina elongata*. Both are calcareous red algae, with a robust skeletal bio-structure that enable them to endure strong wave conditions [39]. The red seaweed *Laurencia papillosa* is also one of the most abundant seaweeds of the Israeli upper tidal zone, occupying the edge of the abrasion platform, and was observed year-round in the current study. It has a unique physical ability to adapt to both light and shade, enabling it to thrive under changing irradiance levels, and was described to possess a unique mucus layer filled with a beneficial symbiotic microbiome that enhances its endurance [40].

In contrast to previous works that have described a seaweed seasonal biodiversity trend of two annual high growth seasons, in spring and autumn [35], the current survey suggests a different pattern, with only one growth maximum occurring during springtime. This growth peak diminishes through summer and autumn and then swiftly resumes towards the end of winter. This shift in the local seaweed bloom may be attributed to the impact of regional environmental changes and global warming. It is widely understood that the spring season offers optimal balanced growth conditions for seaweeds to prosper. The abiotic factors of temperature and irradiance are favorable, coupled with a high level of nutrients following the winter mixing and the run-off from terrestrial sources. Through this mechanism, nutrients are elevated from the sea bottom layer to the upper water column and sea surface, nourishing the algal flora [41,42]. Interestingly, this seasonal cycle was recently similarly described for the phytoplankton communities in the upper 50 m in the eastern Israeli Mediterranean [27,43]. Finally, the high seaweed growth documented during the winter of 2018 likely occurred due to intense winter storms in this particular year, which generated a plume of coastal freshwater rich in nutrients and submarine groundwater discharge. This phenomenon increased the general autotrophic activity and may have significantly impacted the local macroalgae community, thereby promoting their growth [44].

### 3.2. Protein Content in the Surveyed Seaweeds

Seasonality has been recorded to have a significant effect on seaweed protein content. Generally, seaweed protein is highest during the period of winter–early spring and lowest during summer–early autumn. Such fluctuations are believed to be particularly associated with a rich nutrient supply and nitrogen availability [45]. For example, the protein content of *Palmaria palmata* collected from the French Atlantic coast, exhibited the highest values during the winter and spring months. A similar pattern has also been reported for other green and brown seaweed species [6,46]. In the current work, most of the species presented significantly higher protein levels during the winter. The seaweeds exploit the dissolved nutrients and nitrogen in their environment to generate proteins and build their tissues [47,48]. Additionally, during autumn and winter, certain seaweeds have been reported to store nitrogen (N) and phosphorus (P) as reserve resources to contend with seasonal light and nutrient limitation periods. This is suggested to be an ecological strategy to enhance their competitive ability [49]. However, not all species demonstrate this trend, as observed in the case of *Codium adhaerens* and *Hypnea musciformis* in the current survey, as well as in other species described in previous studies [50].

Generally, green and red seaweeds are reported to have high protein concentrations, comprising approximately 10–30% and 35–47% of their dry weight, respectively, while brown seaweeds typically have lower protein content, averaging 5–15% on a DW basis [6,51]. In the present survey, *Dictyota dichotoma* revealed a relatively high protein concentration, consistent with values close to 14% DW previously reported for this species [52]. *Padina pavonica* and *Sargassum vulgare* also had values within the range of those in earlier studies (9–13.6, 7.7–16.3% DW, respectively) [53,54]. Among the red seaweeds, *Jania rubens* displayed the lowest values (4–6.5% DW) and *H. musciformis* presented the highest protein content (12.7–14% DW), both within the range of previously reported protein values for these species [55,56]. *Laurencia papillosa* protein levels were lower than the 34% DW value described by Kumar and Murugesan (2018) [57]. Among the green seaweeds, *Ulva rigida* mean protein content was higher than the values of 6.6 and 4.6% DW noted in earlier works [58,59]. *C. adhaerens* displayed similar or higher protein levels compared to other *Codium* spp. protein contents of 7 and 6.1% DW reported by McDermid and Stuercke (2003) [60] and Manivannan et al. (2008) [61], respectively. Variations in seaweed protein content in comparison to those in the literature can also be attributed to specimen handling, different drying techniques and evaluation methodologies [7]. In the current study, a nitrogen-to-protein conversion factor of 5 was applied, specifically calculated for seaweed protein nutritional evaluation. This contrasts with the factor of 6.25 used in many previous studies, which has been suggested to be significantly overestimated [62,63].

Despite the encouraging potential of seaweed proteins, there are still aspects that hinder their possible contribution and utilization, including digestibility and the presence of toxic compounds. Future studies will need to focus on novel and viable extraction and pre-treatments methods that could help to overcome these issues, improving seaweed proteins’ safety and bioavailability [64,65,66,67].

### 3.3. Antioxidant Activity of Natural Seaweeds

In the current study seasonality had a significant effect on seaweed antioxidant activity. For most species evaluated, particularly the brown seaweeds, the antioxidant capacity was highest during the spring. A similar trend was observed in work conducted on seaweeds collected from a coastal marine ecosystem located in Alexandria, Egypt, in the eastern Mediterranean [68]. Celis-Plá et al. (2016) [69], similarly found that *Cystoseira tamariscifolia* accumulated a high concentration of antioxidant and phenolic compounds during spring, while in contrast, during summer, photoinhibition and low nutrient levels led to an opposite trend. The spring season in the Israeli Mediterranean Sea may provide ideal conditions for the synthesis of high levels of antioxidant substances in seaweeds. As noted, the nutrient-rich seawater resulting from the preceding winter mixing, followed by increasing irradiance, serve as building blocks and energy supply for the synthesis of the necessary antioxidant metabolites [27,70]. Elevated nutrient levels have already been documented to play a key role in enhancing seaweeds’ antioxidant defenses against abiotic stressors [71,72]. Lesser et al. (2006) [73] demonstrated that nitrogen-supplied algae were more resilient to high ultraviolet radiation (UVR) compared to nitrogen-limited algae, aiding the algal cell to repair radiation damage and increase the turnover of critical proteins, photosynthetic pigment complexes, and necessary antioxidant defense compounds.

In the current survey, *Codium adhaerens* exhibited antioxidant levels similar to those reported by Pedro et al. (2022) [74] for *Codium* sp., 2.22–2.75 mg TE g^−1^ DW. The antioxidant level of *Ulva rigida* was slightly lower than values described elsewhere (2.36–3.55 mg TE g^−1^ DW) [70]. In general, the antioxidant capacity of the studied red and brown seaweeds either aligned with or surpassed that of other seaweed species from the same phyla in previous studies, which presented antioxidant values ranging from 1.2 to 15 mg TE g^−1^ DW [75]. In the current work, the brown species exhibited significantly high antioxidant levels compared to the red and green species. *Sargassum vulgare* in particular exhibited a relatively high antioxidant capacity of 56.98 ± 12.25 mg TE g^−1^ DW. Brown seaweeds have been widely acknowledged for their rich content of antioxidant compounds, including sulfated polysaccharides, phenolic compounds, flavonoids, fucans, fucoxanthin, β-carotene, and more. This indicates their high potential for medical and nutritional antioxidant applications [53,76,77,78]. Antioxidant activity in general may be associated with compounds that provide photoprotection and the neutralization of ROS such as carotenoids and phycobiliproteins pigments, phenolic compounds, MAAs, and others [79].

### 3.4. Phenolic Compounds in the Surveyed Seaweeds

Among the seaweed phyla, the brown seaweeds have been identified as an outstanding and the highest source of phenolic compounds, from simple phenolic acids to more complex polymers such as tannins (mainly phlorotannins). In the present work, brown seaweeds demonstrated the highest relative phenolic content, with *Sargassum vulgare* showing the highest levels. The phenol levels observed were in line with previously reported values for both brown and green seaweeds [79]. Throughout this work, it became evident that the phenolic content of seaweeds may vary significantly due to environmental factors, making chemical variability often complex and difficult to predict [80]. Recent studies have highlighted the significant impact of seasonality on the nutritional and chemical profiles of brown seaweeds [19,81]. It is generally assumed that a higher phenolic content, along with increased nutritional content, including polyunsaturated fatty acids (PUFA), vitamins, and minerals, and an enhanced general bioactivity of brown algae, may occur during the hot and dry summer seasons, associated with higher sea temperatures and irradiance [82,83,84]. For example, Čagalj et al. (2022) [19] found that the highest phenolic content in *Cystoseira compressa* was observed during June. However, other seaweed studies have not indicated a uniform seasonal effect on the concentration of phenolic substances, including in genera such as *Padina*, *Colpomenia*, *Saccharina*, and *Dictyota* [81,85,86]. Phenolic compounds were shown to present a photoprotective role and to increase under climate change scenarios such as elevated CO_2_ and temperature [87].

From the current survey, it appears that the different seaweeds do not exhibit a common seasonality trend in their phenolic content. Each species presented a different response, accumulating phenols in different seasons. For example, *S. vulgare* exhibited the highest phenol levels during spring and *Dictyota dichotoma* reached its highest levels in autumn, while *Padina pavonica* maintained relatively constant phenolic content throughout the year. It is possible that each species employs a different ecological strategy or activates varying levels or forms of phenolic compounds based on its biological needs and life history. Given the diversity of polyphenols, such as phenolic acids, flavonoids, stilbenes, tannins, and lignans, more precise monitoring and chemical evaluation would appear to be necessary.

### 3.5. Mycosporine-like Amino Acids (MAAs) in Israeli Seaweeds: An Untapped Potential

The production of MAAs, nitrogenous photoprotective compounds, largely depends on the availability of nutrients, particularly inorganic nitrogen, and solar irradiance, especially blue wavelengths of photosynthetically active radiation (PAR) and UVR [71,88,89]. The composition and formation of MAA molecules may change according to seaweed habitat or culture conditions, even within the same algal species [70,88,90,91]. Red seaweeds in intertidal zones generally exhibit higher MAA content compared to species in subtidal or deep shaded habitats [90,92,93]. Similarly, seaweeds in low-latitude regions may present higher MAA content compared to those in high-latitude regions [94], positioning the Israeli Mediterranean in the Levantine Basin as a promising region for high MAA occurrence. In the current evaluation, a significant seasonality effect was observed for *Jania rubens*, which presented a significantly higher total MAA content during winter. The other studied red seaweeds maintained constant total MAA levels throughout the year. However, *Hypnea musciformis* presented the MAA Porphyra-334 exclusively during winter. Although higher MAA levels are typically expected in more sunny seasons such as summer and spring, it seems that the Israeli winter too can provide the necessary solar energy for MAA synthesis, accompanied by nutrient-enriched seawater. Previous studies have reported a significant seasonal trend in seaweed MAAs, with levels usually increasing in spring but sometimes in winter, attributed to higher daily irradiance and nitrogen-enriched waters [95,96].

The MAA composition and content evaluated in the current work, characterizing each red seaweed species, generally align with those observed in previous studies [70,97]. The MAA palythine was consistently observed in all three red seaweed species throughout the year, with particularly elevated levels in *J. rubens* during spring and summer. This observation may be associated with the significant antioxidant capacity observed in *J. rubens*, particularly during the hot sunlit seasons. Previous studies have demonstrated that palythine exhibits the highest antioxidant capacity among common red seaweed MAAs and also serves as a multifunctional photoprotective molecule [98,99]. This suggests that the relatively challenging environmental conditions experienced by seaweeds prompt them to utilize palythine as a protective antioxidant agent, indicating a physiological response to stress [70].

An annual investigation of MAAs could provide valuable insights into the environmental factors influencing their synthesis, both at the species level and at the level of the individual MAA molecule. Sun-rich regions like the Israeli coastline offer a strong potential for the natural production of photoprotective molecules. Such knowledge could help to determine optimal harvesting periods for a high-MAA seaweed biomass in nutrient-enriched coastal areas and serve as a guide for controlled aquaculture cultivation [18,70].

### 3.6. Practical and Future Perspectives

An important next step will be to conduct extensive routine surveys of local marine flora, expanding the analysis to include additional phytochemical compounds and identifying additional seaweed species as potential candidates for aquaculture. Such efforts will support both ecological assessments and the growing industrial demand for sustainable marine resources.

In this study, in line with previous findings, we suggest that red seaweeds, for example, can be particularly well-suited for cosmetic applications due to their natural sunscreen and antioxidant compounds (e.g., MAAs). Brown seaweeds show strong potential for both cosmetic and pharmaceutical use, owing to their high phenolic content and antioxidant capacity. Green and red seaweeds, with their relatively high protein levels, appear promising for functional foods and nutraceutical applications. Ultimately, each seaweed species offers distinct bioactive properties that can be harnessed for specific health-promoting and nutritional uses.

## 4. Materials and Methods

### 4.1. Seaweed Survey and Sample Collections

A seaweed field survey was conducted at five different sites along the Israeli Mediterranean Sea shoreline, between March 2018 and June 2020 (Figure 1 and Figure 2). During this period, one or two surveys were conducted each month, totaling 30 sampling days, and 320 individual seaweeds were surveyed. Sampling was carried out mainly from the intertidal and shallow subtidal zones of the abrasion platforms. We categorized four seasons, each comprising a three-month period, as follows: spring (March–May), summer (June–August), autumn (September–November), and winter (December–February) (Figure 1 and Figure 2).

#### Taxonomic Descriptions

Species were described taxonomically mainly following visual and microscopic observations. Literature descriptions and herbarium specimens were used as taxonomical keys for comparison and identification (https://www.seaweedherbarium.com, accessed on 1 May 2025). Prior to identification, species were pretreated as detailed below. Specimens that could not be positively identified to the species level were labeled as sp. and maintained for further genetic analysis (Table 1, Appendix A).

### 4.2. Chemical Analysis Sample Preparation

Eight dominant, representative species were evaluated for seasonal changes in their chemical constitutes: two greens, *Ulva rigida* and *Codium adhaerens*, three reds, *Jania rubens*, *Laurencia papillosa* and *Hypnea musciformis*, and three brown seaweeds, *Dictyota dichotoma*, *Padina pavonica* and *Sargassum vulgare* (Figure 3). Freshly collected seaweed material was carefully washed with tap water to discard salt, debris, and epiphytes, and finally centrifuged with a kitchen salad spinner to remove excess water. Samples were then freeze-dried using a lyophilizer (Christ, Alpha 1–2 LD plus, Osterode am Harz, Germany), grounded to a fine homogenized powder and stored at −20 °C prior to further chemical analyses. Triplicates from each sample were taken for the different chemical analyses. All seaweed samples were also photographed and documented (Figure 3, Appendix A).

### 4.3. Protein Content Evaluation

Protein content (% of DW) was evaluated using Kjeldahl nitrogen analysis [100]. Dried samples were evaluated for nitrogen and converted to total protein content using a corrected nitrogen-to-protein factor of five, as proposed by Angell et al. (2016) [62].

### 4.4. Determination of Antioxidant Activity

The seaweed antioxidant activity was evaluated using the ABTS method [101]. Seaweed samples were first extracted by adding 1 mL of DDW to 20–50 mg of dry seaweed powder. The samples were vortexed, placed in an ultrasonic bath for 30 min, and then remained overnight in darkness at 25 °C. On the following day, extracts were centrifuged, and supernatants were used for analyses. ABTS reagent was prepared in sodium phosphate buffer (0.1 M, pH 6.5), using ABTS (2,2-azino-bis (3-ethylbenzothiazoline-6-sulphonic acid, 7 mM) and potassium persulfate (K_2_S_2_O_8_, 2.45 mM). The reagent was incubated in darkness at room temperature for 12–16 h, allowing complete formation of the radical. The assay reaction was performed by adding 950 µL of diluted ABTS reagent with 50 µL seaweed sample extract. The samples were agitated, and absorbance was read via an Agilent Cary 60 UV-Vis Spectrophotometer (Santa Clara, CA, USA) at 727 nm after 8 min of incubation. The blank was phosphate buffer. Antioxidant activity was calculated using the following formula:

$$\text{AA\%}=\left[{\mathrm{OD}}_{i}-{\mathrm{OD}}_{f}/{\mathrm{OD}}_{i}\right]\times 100$$
where OD_i_ is the initial optical density absorbance, and OD_f_ is the final optical density absorbance. Quantification of antioxidant compounds was determined using a standard curve with different Trolox (6-hydroxy-2,5,7,8-tetramethylchroman-2-carboxylic acid) concentrations. The results were expressed as mg of Trolox Equivalents (TE) per g of seaweed dry weight (mg TE g^−1^ DW).

### 4.5. Determination of Phenolic Content

Quantification of phenolic compounds was performed according to the Folin–Ciocalteu method [102] with some modifications. Seaweed samples were first extracted by adding 1 mL of phosphate buffer (0.1 M, pH = 6.5) to 20 mg of dry seaweed powder. The samples were vortexed and remained overnight in darkness at 4 °C. The reaction was performed by adding 100 µL of each seaweed extract to 700 µL distilled water, 50 µL of the Folin–Ciocalteu reagent, and, finally, 150 µL 20% anhydrous sodium carbonate (Na_2_CO_3_). The solution was vortexed and incubated at 4 °C in darkness for 2 h. Absorbance was measured at 760 nm using a UV-visible spectrophotometer (UV-2700i Shimadzu, Duisburg, Germany). The blank included all reagents, and the crude extract was replaced by distilled water. Phenolic content was evaluated by constructing a standard curve using different phloroglucinol concentrations. Results were expressed as mg of phloroglucinol equivalent (PE) per g of seaweed dry weight (mg PE g^−1^ DW).

### 4.6. Analysis of Mycosporine-like Amino Acids (MAAs)

MAAs were assayed according to Korbee–Peinado et al. (2004) [103]. An amount of 50 mg of dried seaweed was incubated in 20% methanol (1 mL) in a water bath at 45 °C for 2 h. Then, 700 μL of the supernatant was taken and evaporated under vacuum at 45 °C (SpeedVac SPD210 Vacuum Concentrator, Thermo Scientific, Waltham, MA, USA). Dried extracts were redissolved in 700 μL of 100% methanol and vortexed for 30 s. After passing through a 0.2-μm membrane filter, samples were analyzed with an Agilent UHPLC system (1260 Agilent InfinityLab Series, Santa Clara, CA, USA) by using a Luna C8 column (Phenomenex) with an isocratic flow of 0.5 mL min and mobile phase of 1.5% methanol and 0.15% acetic acid in ultrapure water. Isolated MAAs through HPCCC [104] were used as standards. Results were expressed as (mg g^−1^ DW).

### 4.7. Statistical Analysis

Statistical analyses were performed using the R statistic program, version 4.3.1, Vienna, Austria. One-way ANOVA (α = 0.05) was used to compare parameters between experiments. Tukey’s HSD test was used for post hoc pairwise comparisons. Data were tested for normality (Shapiro–Wilk test) and homogeneity of variance (Levene’s test). When ANOVA assumptions were not met, a permutation ANOVA test of 5000 repetitions was used, and a Games–Howell test was applied for post hoc comparison. Data in tables and figures are expressed as mean ± SD.

The complete statistical data are summarized in Appendix A. In the results chapter, when the results of multiple tests are mentioned together, they are referred to Appendix A.

## 5. Conclusions

Belonging to ancient and distinct evolutionary dynasties, seaweeds have evolved unique molecules that enable them to thrive in challenging fluctuating environments. They contribute to nearly 3000 different natural products, encompassing around 20% of the entire chemistry of the marine realm [105]. From a biotechnological perspective, seaweeds may be defined as ‘biological machines’, able to produce valuable nutritional and bioactive materials that can be harnessed for human benefit [1,51,106]. This study focused on natural seaweed communities along the Israeli Mediterranean Sea, specifically within the Levantine Basin, the easternmost and one of the most ecologically distinctive regions of the Mediterranean Sea [25,31]. As discussed throughout this work, environmental variability may stimulate the synthesis of protective substances and secondary metabolites in seaweeds. The concentrations of active and beneficial compounds measured in the surveyed species were generally within or above the known range for their genera. Notably, certain species, particularly brown seaweeds, exhibited especially high levels of antioxidants and phenolic compounds.

We suggest that this region may function as a unique ‘incubator habitat’ serving as a hotspot for adapted seaweed species. It may offer insight into potential climate change scenarios and help to predict or ‘foster’ resilient seaweed species capable of enduring future global warming, potentially becoming dominant in other regions under such conditions [42,107]. Those species are likely to be highly tolerant and will possess the necessary physiological, structural, and chemical adaptations, including enrichment in distinctive and beneficial bioactive compounds [108,109,110]. In the current survey we identified the dominant species and highlighted some of their chemical advantages. Our findings indicate how the different seasons, characterized by various environmental factors, may affect the seaweeds’ chemistry. Such knowledge can be translated into aquaculture cultivation protocols to define optimal growth conditions for different seaweed species, enabling the enhancement of their beneficial metabolites for use in nutrition and health applications, including superfoods, dietary supplements, cosmetics, nutraceuticals, and pharmaceuticals [48,70]. Beyond the seaweeds’ practical advantages, their primary benefit lies in their environmental attributes. Cultivating seaweed helps to conserve and balance the natural marine ecosystems. To address environmental degradation, humanity must undergo a paradigm shift and rely on science-based solutions and sustainable technologies [111,112]. Consequently, seaweeds can be defined as a sustainable resource that, if utilized correctly will assist humanity in tackling ongoing global challenges such as the climate crisis, food security, pollution, resource depletion, and the crossing of planetary boundaries.

## Figures and Tables

**Figure 1 marinedrugs-23-00320-f001:**
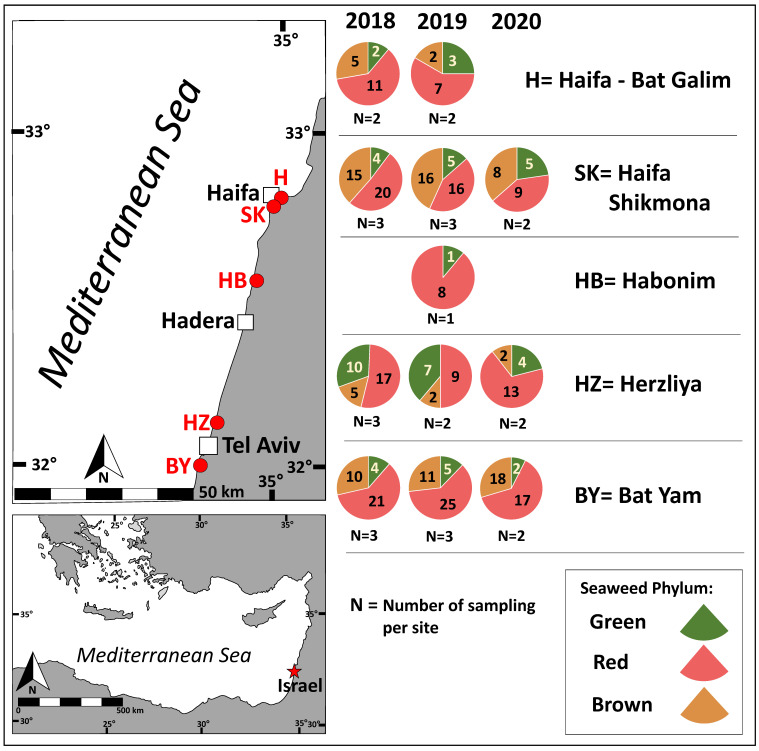
Seaweed phyla abundance according to sampling site. Chlorophyta (green), Heterokontophyta class Phaeophyceae (brown), and Rhodophyta (red). Sampling sites are labeled in red on the map. The number of species sampled per phylum is indicated (each species was counted only once), along with the sampling effort per site (N).

**Figure 2 marinedrugs-23-00320-f002:**
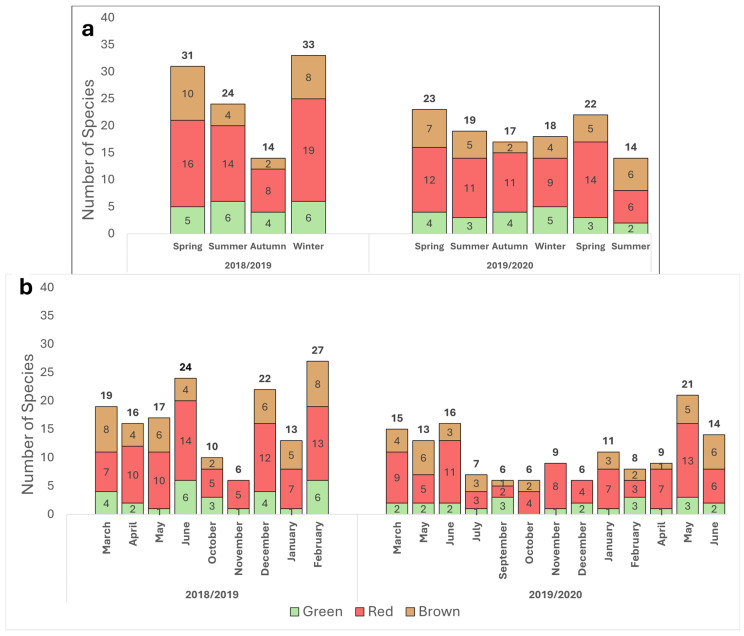
Seasonal (**a**) and monthly (**b**) distribution of seaweeds by phyla, from March 2018 through June 2020. Chlorophyta (green), Heterokontophyta class Phaeophyceae (brown) and Rhodophyta (red). The number of species per phylum is indicated inside each column, and total number of species recorded is noted at the top. Each species was counted only once per the defined period.

**Figure 3 marinedrugs-23-00320-f003:**
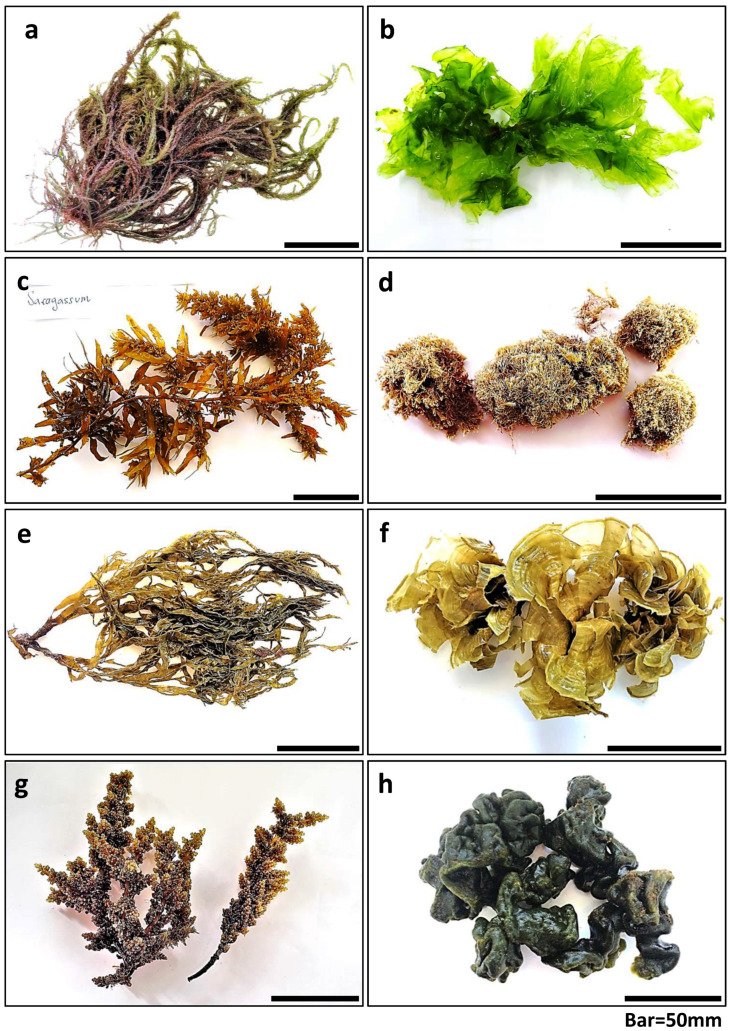
The eight seaweed species that were chemically evaluated: (**a**) *Hypnea musciformis*, (**b**) *Ulva rigida*, (**c**) *Sargassum vulgare*, (**d**) *Jania rubens*, (**e**) *Dictyota dichotoma*, (**f**) *Padina pavonica*, (**g**) *Laurencia papillosa*, and (**h**) *Codium adhaerens*.

**Figure 4 marinedrugs-23-00320-f004:**
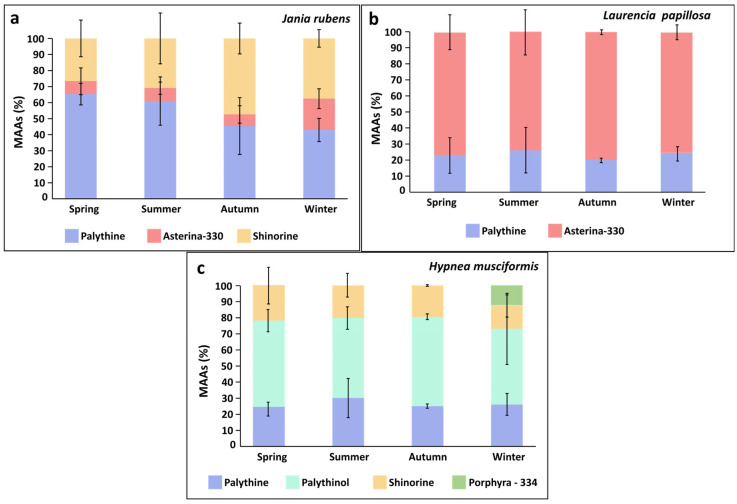
Seasonal average proportions (%) of Mycosporine-like Amino Acids (MAAs) throughout the sampling period for (**a**) *Jania rubens*, (**b**) *Laurencia papillosa*, and (**c**) *Hypnea musciformis*.

**Table 1 marinedrugs-23-00320-t001:** Seaweed species sampled in the survey classified according to phylum. Chlorophyta (green seaweeds), Heterokontophyta, class Phaeophyceae (brown seaweeds), and Rhodophyta (red seaweeds). In total 55 species.

Seaweed Species		
Chlorophyta (12)	Rhodophyta (29)	Heterokontophyta (Phaeophyceae) (14)
*Bryopsis mucosa*	*Gracilaria conferta*	*Stypopodium schimperi*
*Bryopsis plumosa*	*Alsidium corallinum*	*Colpomenia sinuosa*
*Caulerpa mexicana*	*Asparagopsis taxiformis*	*Cystoseira compressa*
*Cladophora pellucida*	*Acanthophora najadiformis*	*Gongolaria rayssiae*
*Cladophora pseudopellucida*	*Chondracanthus acicularis*	*Dictyota* sp.
*Cladophoropsis membranacea*	*Chondria dasyphylla*	*Dictyota dichotoma*
*Codium adhaerens*	*Corallina elongata*	*Ectocarpus siliculosus*
*Codium parvulum*	*Dasya* sp.	*Halopteris scoparia*
*Ulva californica*	*Digenea simplex*	*Lobophora variegata*
*Ulva compressa*	*Galaxaura rugosa*	*Padina pavonica*
*Ulva fasciata*	*Gelidium pusillum*	*Sargassum vulgare*
*Ulva rigida*	*Gelidium crinale*	*Scytosiphon lomentaria*
	*Gracilaria bursa-pastoris*	*Spatoglossum solieri*
	*Gracilaria cornea*	*Taonia atomaria*
	*Gracilaria* sp.	
	*Grateloupia* sp.	
	*Halymenia dichotoma*	
	*Hypnea musciformis*	
	*Hypnea* sp.	
	*Jania rubens*	
	*Laurencia papillosa*	
	*Laurenciella marilzae*	
	*Liagora* sp.	
	*Nemalion helminthoides*	
	*Neopyropia elongata* (formerly *Porphyra leucosticta*)	
	*Pterocladiella capillacea*	
	*Rytiphlaea tinctoria*	
	*Solieria filiformis*	
	*Spyridia filamentosa*	

**Table 2 marinedrugs-23-00320-t002:** Evaluated compounds, including total protein (% of DW), antioxidant activity (mg TE g^−1^ of DW), and polyphenol levels (mg PE g^−1^ of DW) for the eight seaweed species sampled throughout the survey, along with the indications of their seasonal variation trends. ‘NA’ indicates the absence of the species in a given season or insufficient sample material for chemical analysis. A significant increase in compound content compared to other seasons is highlighted in bold.

**Protein (% DW)**				
**Species/Season**	**Spring**	**Summer**	**Autumn**	**Winter**
*Laurencia papillosa*	7.84 ± 1.27	7.07 ± 1.56	9.27 ± 0.05	**12.29 ± 1.23**
*Jania rubens*	4.85 ± 0.62	4.01 ± 1.36	4.81 ± 0.52	**6.48 ± 0.35**
*Hypnea musciformis*	12.77 ± 2.71	12.68 ± 1.12	13.28 ± 0.11	14.07 ± 1.17
*Padina pavonica*	6.83 ± 1.24	7.31 ± 1.23	7.55 ± 0.97	**9.49 ± 2.09**
*Sargassum vulgare*	6.68 ± 2.94	6.17 ± 0.06	NA	**9.67 ± 0.81**
*Dictyota dichotoma*	9.14 ± 0.91	8.07 ± 0.78	10.07 ± 0.15	**14.4 ± 0.57**
*Ulva rigida*	4.84 ± 0.97	5.97 ± 2.7	6.08 ± 0.37	**9.58 ± 1.23**
*Codium adhaerens*	8.35 ± 0.62	7.97 ± 0.59	7.66 ± 0.21	8.18 ± 0.56
**Antioxidants (mg TE g^−1^ DW)**				
**Species/Season**	**Spring**	**Summer**	**Autumn**	**Winter**
*Laurencia papillosa*	7.09 ± 0.62	8.54 ± 0.85	5.89 ± 1.13	7.03 ± 0.89
*Jania rubens*	**1.32 ± 0.33**	1 ± 0.2	1.15 ± 0.28	**1.69 ± 0.43**
*Hypnea musciformis*	2.12 ± 0.54	2.55 ± 1.02	1.71 ± 0.06	2.05 ± 0.24
*Padina pavonica*	**22.12 ± 9.11**	10.97 ± 5.01	13.95 ± 8.07	7.71 ± 2.37
*Sargassum vulgare*	**56.98 ± 12.25**	46.96 ± 3.75	NA	26.87 ± 10.5
*Dictyota dichotoma*	**11.74 ± 5.63**	6.65 ± 2.89	4.99 ± 0.16	6.2 ± 1.62
*Ulva rigida*	1.73 ± 0.63	1.88 ± 0.17	0.47 ± 0.03	1.92 ± 0.3
*Codium adhaerens*	**1.72 ± 0.17**	1.23 ± 0.17	0.99 ± 0.27	1.24 ± 0.17
**Polyphenols (mg PE g^−1^ DW)**				
**Species/Season**	**Spring**	**Summer**	**Autumn**	**Winter**
*Padina pavonica*	7.7 ± 5.61	6.4 ± 1.88	10.73 ± 5.69	6.17 ± 2.68
*Sargassum vulgare*	**36.97 ± 8.43**	NA	NA	20.56 ± 8.17
*Dictyota dichotoma*	8.72 ± 0.34	7.53 ± 3.17	**10.61 ± 0.85**	6.25 ± 0.84
*Ulva rigida*	2.59 ± 0.78	2.11 ± 0.34	0.84 ± 0.25	2.94 ± 0.63
*Codium adhaerens*	2.47 ± 0.2	2.41 ± 0.2	2.37 ± 0.71	1.93 ± 0.39

**Table 3 marinedrugs-23-00320-t003:** Mycosporine-like amino acids (MAAs) average content (mg g^−1^ DW) in *Jania rubens*, *Laurencia papillosa*, and *Hypnea musciformis* sampled across different seasons during the survey. The total MAA content for each species in each season is highlighted in bold. ‘NA’ indicates the absence of a given MAA.

	MAAs (mg g^−1^ DW)				
Species/Season		Spring	Summer	Autumn	Winter
*Jania rubens*	Palythine	0.16 ± 0.04	0.05 ± 0.03	0.1 ± 0.07	0.2 ± 0.05
	Asterina-330	0.05 ± 0.02	0.01 ± 0.01	0.03 ± 0.01	0.09 ± 0.04
	Palythinol	NA	NA	NA	NA
	Shinorine	0.08 ± 0.03	0.04 ± 0.04	0.09 ± 0.01	0.18 ± 0.02
	Porphyra-334	NA	NA	NA	NA
	**Total**	**0.25 ± 0.06**	**0.1 ± 0.07**	**0.21 ± 0.07**	**0.47 ± 0.06**
*Laurencia papillosa*	Palythine	0.06 ± 0.04	0.03 ± 0.02	0.06 ± 0.01	0.07± 0.05
	Asterina-330	0.29 ± 0.21	0.13 ± 0.11	0.22± 0.04	0.22 ± 0.14
	Palythinol	NA	NA	NA	NA
	Shinorine	NA	NA	NA	NA
	Porphyra-334	NA	NA	NA	NA
	**Total**	**0.35 ± 0.24**	**0.15 ± 0.13**	**0.28 ± 0.05**	**0.29 ± 0.18**
*Hypnea musciformis*	Palythine	0.31 ± 0.05	0.39 ± 0.11	0.32 ± 0.07	0.38 ± 0.12
	Asterina-330	NA	NA	NA	NA
	Palythinol	0.72 ± 0.28	0.77 ± 0.39	0.7 ± 0.09	0.81 ± 0.49
	Shinorine	0.28 ± 0.07	0.32 ± 0.15	0.25 ± 0.05	0.25 ± 0.15
	Porphyra-334	NA	NA	NA	0.45 ± 0.04
	**Total**	**1.43 ± 0.41**	**1.47 ± 0.61**	**1.28 ± 0.21**	**1.46 ± 0.55**

## Data Availability

All data are available in the main text or the Appendix A.

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
