# Peer review of "Seaweeds of the Israeli Mediterranean Sea: Nutritional and Biotechnological Potential Through Seasonal and Species Variation"

_marinedrugs, 2025, doi:10.3390/md23080320_

Round 1

Reviewer 1 Report

Comments and Suggestions for Authors

This manuscript seems like a competent contribution to local knowledge of the intertidal marine flora of Isreal. It is not clear to me why it is submitted to Marine Drugs. The authors attempt to say that the harsh intertidal environment there “may foster” species “with distinct chemical properties” they present no basis for that and indeed, most intertidal environments are ‘harsh’ in a variety of ways. This is also of only limited international interest. That said, if the Editor(s) view this as appropriate for the journal I see no reason for it not to be accepted. The taxonomy appears up to date, and within my experience with some but not all of the biochemical methods employed, they are appropriate.

A ‘minor’ but important comment is that all species names should be italicized. The authors are inconsistent with that early in the manuscript and then pretty consistently do not use italics in the latter parts.

Line 139: Why Ulva sp. here? Do you mean Ulva rigida? That is the species focused on elsewhere.

Line 261: Except for parenthetically in line 152, the rest of the manuscript refers t the Heterokontophyta. Why switch to Phaeophyceae here? I actually think Phaeophyceae would be better throughout since the Heterokontophyta includes so many other, widely divergent Classes. Either way, be consistent.

Author Response

Thank you kindly for taking the time to review our manuscript. We greatly appreciate your insightful and constructive comments, which have, in our opinion, significantly improved the quality of the article. Please find below our detailed responses to each of your comments.

Comment 1: The authors attempt to say that the harsh intertidal environment there “may foster” species “with distinct chemical properties” they present no basis for that and indeed, most intertidal environments are ‘harsh’ in a variety of ways. This is also of only limited international interest.
Response 1: Thank you for this important observation. We agree with your comment and acknowledge that our original phrasing was too strong. Accordingly, we have refined the statement to clearly present it as a hypothesis rather than a confirmed conclusion. We now explicitly state that our reasoning is based on the known and well-studied environmental characteristics of the region.
In addition, the terminology throughout the manuscript has been adjusted and softened to reflect a more cautious interpretation.
This clarification has been incorporated into the Abstract (page 1, line 29), supported by relevant literature in the Introduction (page 3, lines 108–114), further discussed and substantiated with references in the Discussion (pages 13,15,16,17 lines 237–254, 347-351, 409-412,434-437), and expanded upon in the Conclusion through an additional explanatory paragraph (page 19, lines 545–553).
We have also added a supporting citation:
Coll, M. et al. (2010). The Biodiversity of the Mediterranean Sea: Estimates, Patterns, and Threats. PLOS ONE, 5, e11842. https://doi.org/10.1371/journal.pone.0011842.

Comment 2: A ‘minor’ but important comment is that all species names should be italicized. The authors are inconsistent with that early in the manuscript and then pretty consistently do not use italics in the latter parts.
Response 2: We agree. All species names have now been reviewed and consistently italicized throughout the manuscript.

Comment 3: Line 139: Why Ulva sp. here? Do you mean Ulva rigida? That is the species focused on elsewhere.
Response 3: Agreed. We revised the text to "Ulva species" and detailed which species, to provide the necessary clarification (page 5, results chapter, lines: 140-141).

Comment 4: Line 261: Except for parenthetically in line 152, the rest of the manuscript refers the Heterokontophyta. Why switch to Phaeophyceae here? I actually think Phaeophyceae would be better throughout since the Heterokontophyta includes so many other, widely divergent Classes. Either way, be consistent.
Response 4: Thank you for this insightful comment. We agree and revised the terminology accordingly for consistency. The class Phaeophyceae is now explicitly mentioned in the Abstract and further in the main text. Additionally, we adopted the more accessible terminology “green, red, and brown seaweeds” throughout the manuscript to enhance clarity and avoid confusion related to broader taxonomic classifications. These changes appear on pages 1,4, 5–7 (lines 22, 125, 129,145, 149–155) and throughout the text.

Reviewer 2 Report

Comments and Suggestions for Authors

The article submitted for review had two main objectives: the first was to study the biodiversity and seasonal changes of algae along the Israeli Mediterranean coast. The second was to investigate the seasonal changes of some chemical components including proteins, polyphenols, MAA and antioxidant levels for locally collected algae over time. Since Marina Drugs publishes scientific articles devoted to the discovery, development, exploitation and production of biologically and therapeutically active compounds from marine habitats, one may question the relevance of the article under review to the scope of the journal. The species diversity of Mediterranean algae is a topic for research in botanical journals, in particular Phycology. It is impossible to claim that among the parameters of the chemical composition there is at least one biologically active substance: the total protein content was determined by the Kjeldahl method, the total polyphenol content was determined by the Folin-Ciocolteu method, the antioxidant activity is characteristic of all plants. Perhaps the authors should have paid more attention to mycosporin amino acids? Primary data on the determination of mycosporin amino acids (chromatograms, standards, etc.) are missing. The authors can be recommended to rewrite the article to meet the requirements of the journal and strengthen the section on the analysis of mycosporin acids with a simultaneous demonstration of their activity on in vitro or in vivo models.
General recommendations
1. Specify which of the authors was involved in the taxonomic analysis of the algae samples. Where are the herbarium samples stored?
2. The introduction should justify interest in specific indicators of the chemical composition from the point of view of their activity. Components that the authors did not determine (α-tocopherol (vitamin E), ascorbic acid (vitamin C), carotenoids, phycobiliproteins, and ROS-scavenging enzyme) should not be listed.
3. In the conclusion and in the abstract, the authors write that special trends and distinctive substances were found for the studied algae due to the extreme conditions of the Israeli Mediterranean Sea. It would be nice if this conclusion followed from the text of the article. So far this is an unfounded statement.
4. It is recommended to write the species name of the algae in italics. It is also necessary to check the correctness of the spelling of the species and generic names of the algae, as errors occur.
5. The authors have done a lot of work to assess the species diversity of the Mediterranean algae, but the practical value of most species is low, especially in terms of use as drugs of marine origin. Species should be divided according to their possible practical significance.

Author Response

Thank you kindly for taking the time to review our manuscript. We greatly appreciate your insightful and constructive comments, which have, in our opinion, significantly improved the quality of the article. Please find below our detailed responses to each of your comments.

Comment 1: Specify which of the authors was involved in the taxonomic analysis of the algae samples. Where are the herbarium samples stored?
Response 1: Thank you for the comment. The authors responsible for analysis, validation, and investigation are listed in the Author Contributions section (page 20, lines 578–583).
Regarding the herbarium, while the samples were not stored as herbarium specimens, they were preserved as described in the Sample Preparation subsection of the Materials and Methods. Indeed the taxonomic identification was conducted with reference to our local herbarium collection (https://www.seaweedherbarium.com), as described in the Taxonomic Descriptions paragraph (page 17, lines 461–465).

Comment 2: The introduction should justify interest in specific indicators of the chemical composition from the point of view of their activity. Components that the authors did not determine (α-tocopherol (vitamin E), ascorbic acid (vitamin C), carotenoids, phycobiliproteins, and ROS-scavenging enzyme) should not be listed.
Response 2: Thank you for this valuable comment. In the Introduction, we initially aimed to provide a broad overview of the known beneficial properties of seaweeds, also aimed for the general reader audience, and therefore included general examples of beneficial seaweed-derived compounds. However, as the Introduction progresses, we gradually narrow the focus toward our specific research goals. We explain and elaborate on the particular compounds analyzed in our study, namely proteins, antioxidants, phenolic compounds, and mycosporine-like amino acids (MAAs) (pages 2,3, lines 62–67, 68–94, and 116–118).

Comment 3:  In the conclusion and in the abstract, the authors write that special trends and distinctive substances were found for the studied algae due to the extreme conditions of the Israeli Mediterranean Sea. It would be nice if this conclusion followed from the text of the article. So far this is an unfounded statement.
Response 3: Thank you for this important observation. We agree with your comment and acknowledge that our original phrasing was too strong. Accordingly, we have refined the statement to clearly present it as a hypothesis rather than a confirmed conclusion. We now explicitly state that our reasoning is based on the known and well-studied environmental characteristics of the region.
In addition, the terminology throughout the manuscript has been adjusted and softened to reflect a more cautious interpretation.
This clarification has been incorporated into the Abstract (page 1, line 29), supported by relevant literature in the Introduction (page 3, lines 108–114), further discussed and substantiated with references in the Discussion (pages 13,15,16,17 lines 237–254, 347-351, 409-412,434-437), and expanded upon in the Conclusion through an additional explanatory paragraph (page 19, lines 545–553).
We have also added a supporting citation:
Coll, M. et al. (2010). The Biodiversity of the Mediterranean Sea: Estimates, Patterns, and Threats. PLOS ONE, 5, e11842. https://doi.org/10.1371/journal.pone.0011842.

Comment 4: It is recommended to write the species name of the algae in italics. It is also necessary to check the correctness of the spelling of the species and generic names of the algae, as errors occur.
Response 4: We agree. All species and genus names have been italicized and checked for spelling accuracy throughout the manuscript.

Comment 5:  The authors have done a lot of work to assess the species diversity of the Mediterranean algae, but the practical value of most species is low, especially in terms of use as drugs of marine origin. Species should be divided according to their possible practical significance.
Response 5: Thank you for this excellent suggestion. We have accordingly added a new paragraph to the Discussion to elaborate on the potential practical significance of the studied seaweed species and their relevance to various industries (page 17, lines 438–451).

Reviewer 3 Report

Comments and Suggestions for Authors

  • This manuscript addresses the diversity and characterization of macroalgae. However, a major revision is required to reconsider as follows.
  • Abstract: should include the potential or application of this collection.
  • Keywords: should be added only for significant keywords for the article.
  • All tables should remove the arrow, which could be present and explain only in the text.
  • Please improve the resolution of all figures to high quality.
  • Please explain the management of the collection seasons, specifically why spring–winter 2018-2019 and 2019-2020 are included. How about spring–summer in the other two columns? Also, add the range of months for each season.
  • Table S2. Should be removed in the supplemented files.
  • A thorough proofreading of the manuscript is essential to address any language, phrasing, style, or academic writing issues that may detract from its overall quality.

Author Response

Thank you kindly for taking the time to review our manuscript. We greatly appreciate your insightful and constructive comments, which have, in our opinion, significantly improved the quality of the article. Please find below our detailed responses to each of your comments.

Comment 1:    Abstract: should include the potential or application of this collection.
Response 1: Thank you for your comment. The potential applications and scientific relevance of the studied seaweeds are included in the Abstract (page 1, lines 30–32), where we emphasized their biotechnological interest and utility.

Comment 2:    Keywords: should be added only for significant keywords for the article.
Response 2: Thank you for the comment, we believe that that keywords provided are relevant, however we replaced “sustainable-bioproducts” with the broader term “sustainability” to increase relevance and searchability (page 1, line 35).

Comment 3: All tables should remove the arrow, which could be present and explain only in the text.
Response 3: Agreed. All arrows have been removed from the tables, and significant values are now highlighted in bold for clarity (pages 10,11, Tables 2 & 3).

Comment 4: Please improve the resolution of all figures to high quality.
Response 4: We appreciate the reminder. High-resolution versions of all figures have been submitted to the journal editors.

Comment 5: Please explain the management of the collection seasons, specifically why spring–winter 2018-2019 and 2019-2020 are included. How about spring–summer in the other two columns? Also, add the range of months for each season.
Response 5: Thank you. We provided a detailed explanation of the seasonal sampling layout and corresponding months in the Materials and Methods section (page 17, lines 453–460). The sampling began in March 2018 and continued for just over two years.

Comment 6: Table S2. Should be removed in the supplemented files
Response 6: We understand your concern; however, we believe from our experience that Table S2 may be valuable for researchers who wish to examine the detailed statistical outputs. We are happy to leave this decision to the editor's decision.

Comment 7: A thorough proofreading of the manuscript is essential to address any language, phrasing, style, or academic writing issues that may detract from its overall quality.
Response 7: We appreciate your very sincere feedback. Following your suggestion, the manuscript has been sent for further professional academic proofreading and revised to ensure improved language, style, and clarity.

Round 2

Reviewer 2 Report

Comments and Suggestions for Authors

The authors have taken into account the comments on the article. Since the academic editor believes that the article can be accepted for consideration, I have no objection to its acceptance.